# Dna2 nuclease-helicase structure, mechanism and regulation by Rpa

**Chun Zhou[1], Sergei Pourmal[1], Nikola P Pavletich[2]\***

[1]Structural Biology Program, Memorial Sloan Kettering Cancer Center, New York, United States; [2]Howard Hughes Medical Institute, Memorial Sloan-Kettering Cancer Center, New York, United States

**Abstract** The Dna2 nuclease-helicase maintains genomic integrity by processing DNA double-strand breaks, Okazaki fragments and stalled replication forks. Dna2 requires ssDNA ends, and is dependent on the ssDNA-binding protein Rpa, which controls cleavage polarity. Here we present the 2.3 Å structure of intact mouse Dna2 bound to a 15-nucleotide ssDNA. The nuclease active site is embedded in a long, narrow tunnel through which the DNA has to thread. The helicase domain is required for DNA binding but not threading. We also present the structure of a flexibly-tethered Dna2-Rpa interaction that recruits Dna2 to Rpa-coated DNA. We establish that a second Dna2-Rpa interaction is mutually exclusive with Rpa-DNA interactions and mediates the displacement of Rpa from ssDNA. This interaction occurs at the nuclease tunnel entrance and the 5′ end of the Rpa-DNA complex. Hence, it only displaces Rpa from the 5′ but not 3′ end, explaining how Rpa regulates cleavage polarity.

## Introduction

Dna2 has nuclease and helicase activities and plays key roles in maintaining genomic integrity. It is involved in the nucleolytic processing of 5′ flaps during Okazaki fragment maturation, of DNA double-strand breaks (DSBs) during homologous-recombination mediated repair, and of stalled replication forks in the intra-S-phase checkpoint (*Bae et al., 2001*; *Cejka et al., 2010*; *Hu et al., 2012*; *Nimonkar et al., 2011*; *Zhu et al., 2008*).

Dna2 was first identified as a replication mutant required for viability in yeast, and was subsequently shown to be involved in trimming long 5′ RNA-DNA flaps from Okazaki fragments during replication (*Budd et al., 1995*). Most flaps are cleaved by Fen1 concomitant with their generation by strand displacement during Pol δ synthesis on the lagging strand. Flaps that escape early cleavage, or those extended by the Pif1 helicase, get long enough to be coated by the Replication Protein A (Rpa), which renders them resistant to Fen1 (*Bae et al., 2001*; *Pike et al., 2009*; *Stith et al., 2008*). Dna2, which can displace Rpa from ssDNA (*Stewart et al., 2008*), trims the flap to a length too short for stable Rpa binding, and restores Fen1 processing (*Ayyagari et al., 2003*; *Bae et al., 2001*; *Gloor et al., 2012*). The lethality of Dna2 deletion in yeast is attributed to persistent Rpa-coated flaps, which recruit Ddc2 (ATRIP in metazoa) and activate the Mec1 (ATR in metazoa) DNA-damage checkpoint (*Chen et al., 2013*; *Zhu et al., 2008*).

Long-flap processing by Dna2 is dependent on Rpa removing ssDNA secondary structure (*Stewart et al., 2008*), an essential Rpa function in many other aspects of DNA metabolism (*Chen et al., 2013*; *Fanning et al., 2006*; *Symington and Gautier, 2011*). Also paralleling other Rpa-dependent processes, the ability of Dna2 to act on Rpa-coated ssDNA is dependent on direct Dna2-Rpa interactions (*Bae et al., 2003, 2001*). The yeast Dna2△405N mutation that reduces Rpa binding also reduces 5′ flap cleavage and DSB resection in vitro, and renders yeast temperature-sensitive for growth (*Bae et al., 2003, 2001*; *Niu et al., 2010*).

**\*For correspondence:** pavletin@
mskcc.org

**Competing interests:** The authors declare that no competing interests exist.

**eLife digest** DNA carries the genetic information that is essential for organisms to survive and reproduce. It is made of two strands that twist together to form a double helix. However, these strands can be damaged when the DNA is copied before a cell divides, or by exposure to radiation or hazardous chemicals. To prevent this damage from causing serious harm to an organism, cells activate processes that rapidly repair the damaged DNA.

"Homologous recombination" is one way in which cells can repair damage that has caused both strands of the DNA to break in a particular place. In the first step, several enzymes trim one of the two DNA strands at each broken end to leave single stranded "tails". Dna2 is one enzyme that is involved in making these tails, but it can only bind to single-stranded DNA so it only acts after another enzyme has made some initial cuts. The exposed single stranded DNA then searches for an intact copy of itself elsewhere in the genome, which promotes its repair. It is important that only one of the two DNA strands is trimmed at each end otherwise the repair will fail. A protein called Rpa is bound to the DNA and is required for Dna2 to correctly trim the DNA. However, it is not clear exactly how Rpa2 regulates Dna2.

Zhou et al. used a technique called X-ray crystallography to analyze the three-dimensional structures of Dna2 when it is bound to single stranded DNA and when it is bound to Rpa. The experiments show that Dna2 adopts a cylindrical shape with a tunnel through which the single-stranded DNA passes through. The region of Dna2 that is capable of trimming DNA – which is called the nuclease domain – is embedded within the tunnel. The entrance to the tunnel is too narrow to allow double-stranded DNA to enter, so this explains why Dna2 can only act on double-stranded DNA that already has a small single-stranded section at the end.

Inside the tunnel, Dna2 displaces Rpa from one of the strands, which allows Dna2 to trim the DNA. However, other molecules of Rpa remain firmly bound to the other strand to protect it from Dna2. These enzymes also act in a similar way to trim DNA before it is copied in preparation for cell division. Zhou et al.'s findings provide an explanation for how Rpa determines which strand of DNA is trimmed by Dna2. Further work is needed to understand how Dna2 and Rpa work with other enzymes to trim DNA.

In DSB resection, Dna2 acts redundantly with Exo1 (*Gravel et al., 2008*; *Mimitou and Symington, 2008*; *Zhu et al., 2008*). Resection of the 5′ terminated DNA strand results in a long track of 3′ overhang ssDNA, which forms a nucleoprotein filament with the Rad51 strand-exchange protein and initiates homologous recombination (*Symington and Gautier, 2011*). In vitro, Dna2, Rpa and the helicase Sgs1 (BLM in mammals) constitute the minimal complex that can carry out long-range resection. Resection is dependent on the nuclease activity of Dna2 and the helicase activity of Sgs1/BLM. Rpa is essential for supporting the helicase activity of Sgs1/BLM in part by sequestering the unwound strands, and also for regulating Dna2 by blocking its 3′ to 5′ exonuclease activity (*Cejka et al., 2010*; *Nimonkar et al., 2011*; *Niu et al., 2010*). In cells, Rpa depletion eliminates long-range DSB resection, and also causes the loss or inappropriate annealing of short 3′ ends generated by Mre11 (*Chen et al., 2013*).

In addition to these functions, Dna2 is implicated in preventing the regression of stalled replication forks, which otherwise can generate aberrant structures resembling recombination intermediates and lead to genomic instability (*Hu et al., 2012*). This is dependent on the Dna2 nuclease activity, consistent with the ability of Dna2 to cleave fork structures with regressed leading or lagging nascent strands in vitro (*Hu et al., 2012*).

Dna2 contains a PD-(D/E)XK superfamily nuclease motif (*Budd et al., 2000*) and a 5′ to 3′ helicase domain (*Bae and Seo, 2000*). It is a ssDNA endonuclease that requires a free end for cleavage, and does not cleave dsDNA, single-stranded gaps, D-loops or RNA (*Bae and Seo, 2000*; *Kao et al., 2004*). In vitro, isolated Dna2 cleaves ssDNA starting at either end, with multiple rounds of incision degrading ssDNA in both the 5′ to 3′ and 3′ to 5′ directions (*Bae and Seo, 2000*; *Masuda-Sasa et al., 2006*). With 5′ flap DNA, cleavage starts ~~10 nucleotides (nts) from the ssDNA end and continues to within ~5 nts of the duplex (*Bae et al., 2001*; *Bae and Seo, 2000*; *Cejka et al., 2010*;

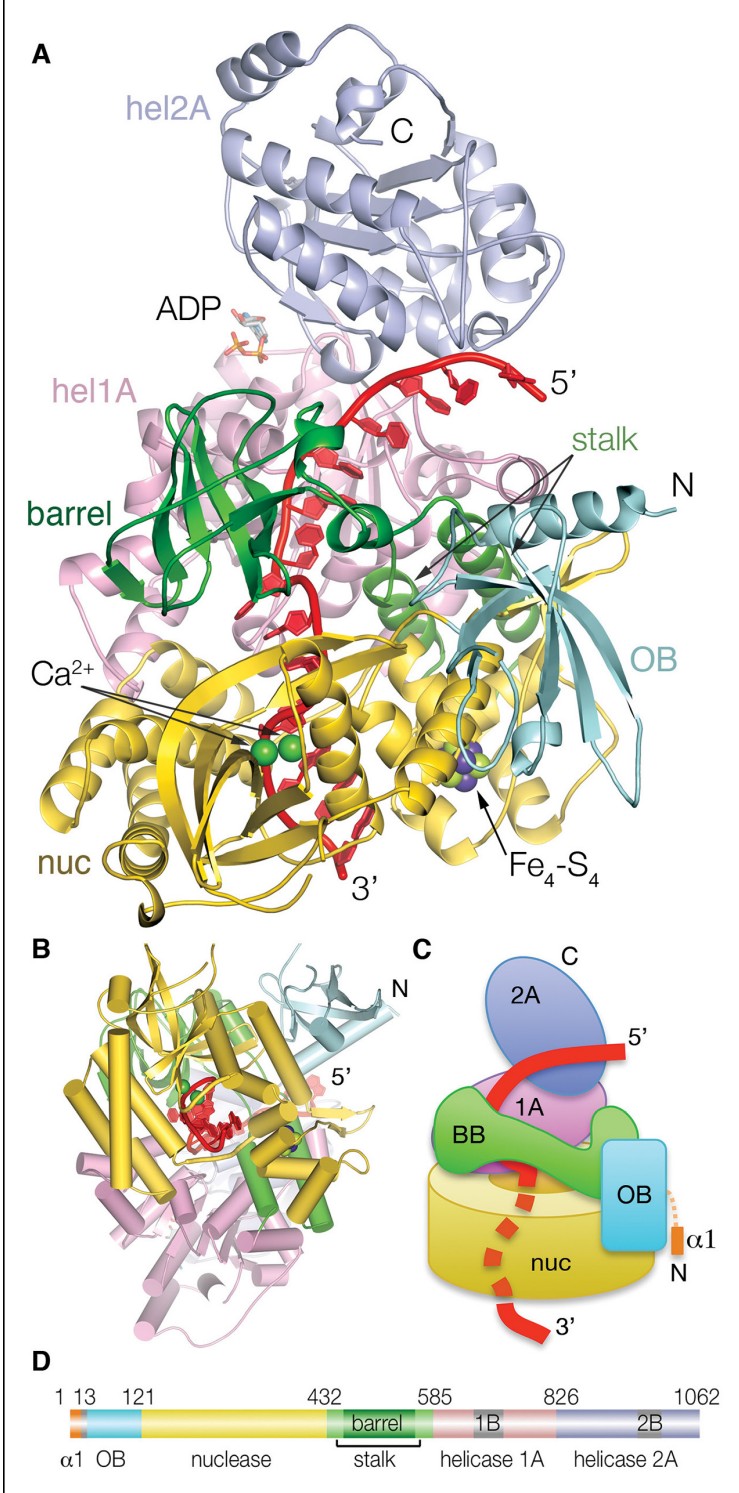

**Figure 1.** Structure of the Dna2-ssDNA complex. (**A**) Cartoon representation of the Dna2-ssDNA complex. The α1 helix, which packs with the hel2A domain of a crystallographic symmetry related protomer, is omitted from this view. The individual domains of Dna2 are colored separately as indicated in (**D**), ssDNA is red, ADP is shown as sticks, calcium ions are shown as green spheres, the iron-sulfur cluster is shown in a space-filling representation (nuc: nuclease domain; hel1A: helicase 1A domain, hel2A: helicase 2A domain, OB: oligonucleotide/oligosaccharide binding domain). (**B**) View looking up the vertical axis of (**A**). (**C**) Schematic of the complex

*Figure 1. continued on next page*

*Figure 1. Continued*

illustrating the relative arrangement of the Dna2 domains, and highlighting its cylinder-like shape. Colored as in (A). (**D**) Linear representation of the Dna2 domains and their boundaries; colored as in (A).

The following figure supplements are available for Figure 1:

**Figure supplement 1.** Dna2 secondary structure and sequence conservation.
**Figure supplement 2.** Dna2 inter-domain interfaces.

---

*Gloor et al., 2012*; *Masuda-Sasa et al., 2006*). It has been suggested that Dna2 loads at the free 5' end of the flap and tracks in the 5' to 3' direction (*Kao et al., 2004*).

The helicase and ATPase activities of Dna2 are substantially weaker than those of other helicases (*Budd et al., 2000*; *Masuda-Sasa et al., 2006*). The helicase activity can be increased by a high ATP to $Mg^{2+}$ ratio, but this also inhibits the nuclease activity through ATP sequestering $Mg^{2+}$ (*Bae and Seo, 2000*; *Masuda-Sasa et al., 2006*). The 5' to 3' polarity of the helicase translocation could, in principle, drive the tracking of Dna2 along the 5' flap. However, ATPase mutations have a minimal effect on 5' flap processing and DSB resection in vitro (*Cejka et al., 2010*; *Niu et al., 2010*; *Zhu et al., 2008*). And, in vivo, ATPase-inactive yeast Dna2 mutants are viable, although they exhibit impaired growth (*Budd et al., 2000*).

To understand the mechanism of action and regulation of this multi-faceted enzyme, we first determined the structure of the intact Dna2-ssDNA complex. The structure revealed that the ssDNA has to thread through a tunnel to bind to Dna2, with a polarity that precludes the cooperation of the helicase and nuclease activities. The requirement for DNA threading prompted us to investigate how Dna2 gains access to Rpa-coated DNA. We provide the structure of a complex between a Dna2 α helix and the Rpa70 OBN domain, both of which are flexibly-tethered and likely serve to recruit Dna2 to Rpa. We also establish a second Dna2-Rpa interaction that helps to displace Rpa from the 5' DNA end, explaining how Rpa restricts the cleavage polarity of Dna2.

## Results and discussion

### Overall structure of the Dna2-ssDNA complex

We determined the 2.3 Å crystal structure of full-length mouse Dna2 (residues 1 to 1062) bound to a ssDNA substrate of 21 nucleotides (nts), 15 of which are well ordered (*Figure 1A* and *Table 1*). The structure also contains an $Fe_4$-$S_4$ iron-sulfur cluster, ADP and two active-site $Ca^{2+}$ ions, which do not support nuclease activity but can mimic magnesium coordination (*Yang et al., 2006*).

The structure consists of a ~310 residue domain that contains the PD-(D/E)XK nuclease motif, followed by a ~450 residue, C-terminal helicase domain that has two RecA-like folds characteristic of the SF1 helicase subfamily (domains 1A and 2A; *Figure 1A,B*). In addition, the structure reveals an OB (oligonucleotide/oligosaccharide-binding) fold domain N-terminal to the nuclease domain, and a β barrel domain that occurs between the nuclease and helicase domains and which is held in place by a stalk of two long alpha helices (*Figure 1* and *Figure 1—figure supplement 1A*).

The overall structure has a cylindrical shape with a central tunnel through which the ssDNA threads (*Figure 1A,B*). The base of the cylinder is formed by the nuclease domain, which adopts a doughnut-like structure with the active site embedded in the central tunnel (*Figure 1C*). The β barrel and helicase 1A domains pack on top of the nuclease doughnut and extend the cylinder and central tunnel. The helicase 2A domain, which packs with the 1A domain as in the ADP states of other SF1 helicase structures, hangs over the tunnel opening at the top of the cylinder. Most of the DNA-binding sites of the nuclease and helicase 1A domains are inside the tunnel, whereas those of the helicase 2A domain are solvent exposed. The OB domain decorates the exterior of the nuclease domain, and is uninvolved in DNA binding. The nuclease domain is the hub that organizes the overall structure. It packs with the flanking OB and β barrel-stalk domains, as well as the helicase 1B domain (*Figure 1C* and *Figure 1—figure supplement 2*).

**Table 1.** Data collection and refinement statistics.

| Data Set | Dna2-5' overhang DNA* | Dna2-ssDNA | apo Dna2 | apo Dna2 (SeMet) | DNA2 $\alpha$1-RPA 70N |
|---|---|---|---|---|---|
| Space group | P22$_1$2$_1$ | P2$_1$2$_1$2$_1$ | P2$_1$2$_1$2$_1$ | P2$_1$2$_1$2$_1$ | C$_1$2$_1$ |
| $a, b, c$ (Å) | 87.2, 118.5, 149.3 | 120.2, 149.2, 172.9 | 120.9, 148.6, 170.5 | 120.9, 148.6, 170.5 | 134.3, 50.9, 76.5 |
| $\alpha, \beta, \gamma$ (°) | 90.0, 90.0, 90.0 | 90.0, 90.0, 90.0 | 90.0, 90.0, 90.0 | 90.0, 90.0, 90.0 | 90.0, 103.9, 90.0 |
| Resolution (Å) | 50.0 - 2.35 (2.43 - 2.35) | 30.0 – 3.11 (3.23 - 3.11) | 50.0 – 3.0 (3.11 - 3.0) | 30.0 – 3.4 (3.52 - 3.4) | 60.0 – 1.5 (1.55 - 1.5) |
| $R_{sym}$ | 12.6 (65.6) | 13.7 (87.5) | 13.1 (78.7) | 15.8 (59.3) | 7.4 (80.1) |
| $R_{pim}$ | 6.3 (36.6) | 6.6 (57.1) | 6.8 (41.9) | 4.0 (15.1) | 5.1 (54.1) |
| I/$\sigma$(I) | 16.4 (1.9) | 13.2 (1.4) | 7.9 (1.3) | 19.3 (4.7) | 23.5 (3.2) |
| Completeness (%) | 99.0 (98.6) | 99.0 (99.2) | 98.3 (98.9) | 100.0 (100) | 84.3 (79.7) |
| Redundancy | 5.0 (3.9) | 6.1 (6.0) | 4.4 (4.5) | 15.1 (15.3) | 2.7 (2.7) |
| Refinement | | | | | |
| Resolution (Å) | 50.0-2.36 | 30.0–3.15 | 50.0–3.0 | | 30.0-1.55 |
| No. of reflections | 57,871 | 51,122 | 54,107 | | 57,455 |
| $R_{work}$/$R_{free}$ (%) | 20.8/24.6 | 22.3/25.6 | 21.1/24.5 | | 23.2/26.7 |
| Protein atoms | 8,298 | 16,536 | 16,536 | | 3,933 |
| DNA atoms | 290 | 674 | 0 | | 0 |
| Cofactor atoms | 41 | 70 | 70 | | 0 |
| Rmsd bond lengths (Å) | 0.009 | 0.009 | 0.009 | | 0.007 |
| Rmsd bond angles (°) B factors (Å$^2$): protein DNA Ca$^{2+}$ water Wilson | 1.4 65.5 107.6 58.9 49.5 58.4 | 1.4 | 1.46 | | 1.3 |

*Only the ssDNA is ordered. Values in parentheses are for the highest-resolution shell.

The ssDNA is positioned with its 5' end at the helicase domain, and its 3' end at the nuclease domain (*Figure 1A*). A 7-nt segment at the 5' end contacts first the helicase 2A domain outside the tunnel and then the 1A domain inside the tunnel. The DNA crosses over from the helicase to the nuclease domains over the next two nucleotides, which are in the vicinity of the β barrel domain in the middle of the tunnel, but do not make any protein contacts (*Figure 1A*). The subsequent 6-nt segment binds to the nuclease domain. Here, the first three nucleotides are fully enclosed by the tunnel, while the last three nucleotides contact the tunnel opening. The DNA bases stack continuously, except for a base step at the helicase, one at the nuclease and one at the transition between the two domains (*Figure 2C*). Two of the three unstacked base steps are at pyrimidine-pyrimidine pairs, and this may contribute to the DNA binding at a well-defined register. An N-terminal acidic/amphipathic α helix (α1; residues 1 to 13) packs with the helicase domain of a symmetry-related complex. The α1 helix is flexibly tethered to the rest of the protein, as the 6 residues that connect it to the OB domain have no electron density in the crystals. As shown later, this helix is one of the two Rpa-binding elements of Dna2.

## Nuclease structure and active site

As predicted, the Dna2 nuclease domain contains the core αββαβ fold of the PD-(D/E)XK nuclease superfamily (*Aravind et al., 2000*; *Pingoud et al., 2005*). Based on the DALI server, its closest structural homologs are the bacteriophage λ exonuclease and the *E. coli* RecB and *B. subtilis* AddB nucleases, all of which are involved in the resection of DNA ends during homologous recombination

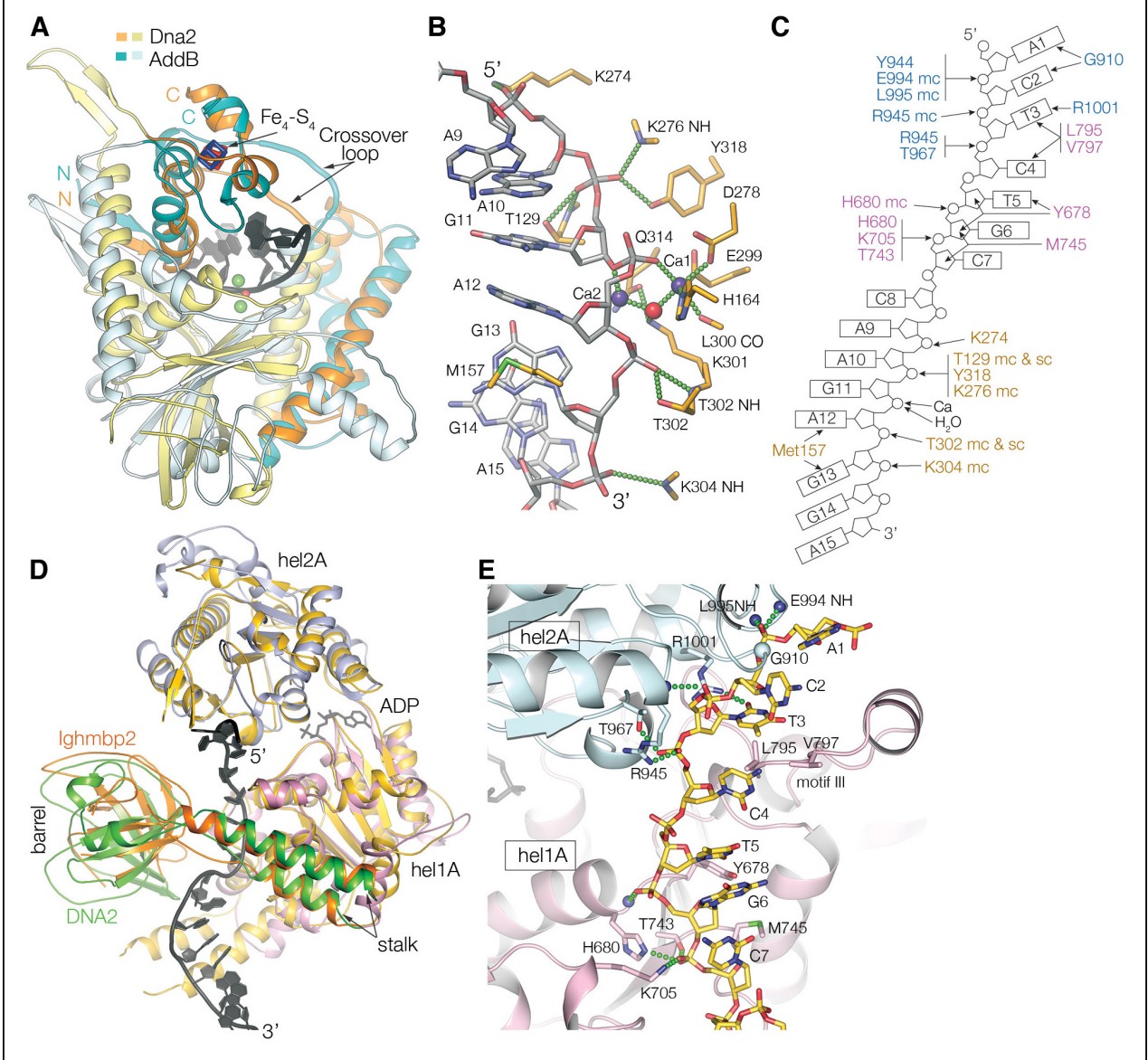

**Figure 2.** Nuclease and helicase domain structures and DNA contacts. (**A**) Superposition of the Dna2 nuclease domain on the AddB nuclease domain. The N-terminal βαα extension (residues 122 to 154) and the C-terminal αα extension (residues 384 to 412) is colored in orange, with the corresponding elements of AddB in dark cyan. Green spheres are calcium ions. (**B**) DNA contacts and active site residues of the Dna2 nuclease domain. Hydrogen bonds are depicted as green dotted lines, calcium ions as blue spheres, water as a red sphere. (**C**) Diagram showing the contacts depicted in (**B**) and (**E**). The residues are colored according to the domain they belong as in *Figure 1D* (mc: main chain, sc: side chain). (**D**) Superposition of the Dna2 helicase domain on Ighmbp2. Dna2 is colored as in *Figure 1D*. The Ighmbnp2 1A (hel1A) and 2A (hel2A) helicase domains are colored gold, its β barrel in light orange, and stalk dark orange. (**E**) DNA contacts of the helicase 1A (pink) and 2A (cyan) domains, showing residues that are involved in either hydrogen bond (green dotted lines) or van der Waals contacts.

The following figure supplements are available for Figure 2:

**Figure supplement 1.** Electron density at the nuclease active site and structural similarity of the Dna2 helicase to Upf1-subfamily RNA/DNA helicases.

(*Krajewski et al., 2014*; *Singleton et al., 2004*; *Zhang et al., 2011*). These superimpose on the ~310-residue Dna2 nuclease domain with a ~2 Å root-mean-square deviation (r.m.s.d.) in the Cα positions of 128, 112 and 106 residues, respectively.

The most extensive similarity is exhibited by AddB, which shares with Dna2 the presence of the iron-sulfur cluster and most of the insertions and extensions that decorate the core fold (*Figure 2A*)

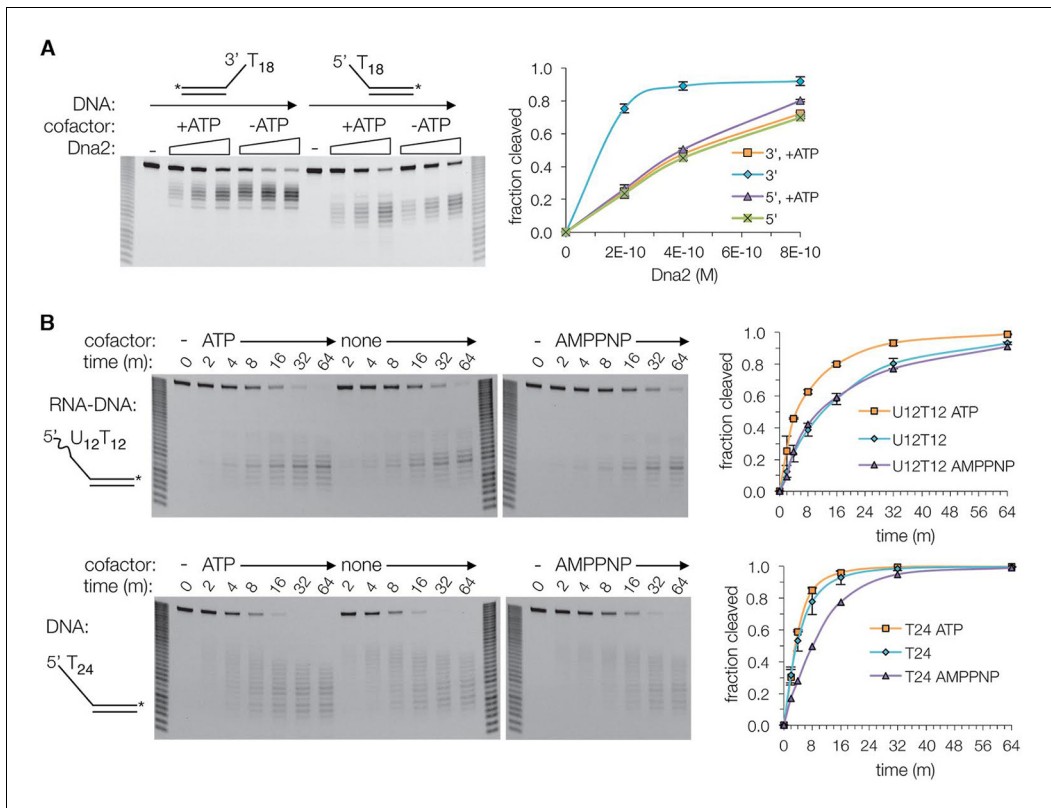

**Figure 3.** Dna2 nuclease activity. (**A**) Denaturing PAGE showing ATP inhibits the nuclease activity on 3' overhang substrate, while slightly increasing it for the 5' overhang substrate. Substrates are at 15 nM. For this and subsequent nuclease assays, cleavage was quantified by loss of substrate and plotted with ± s.d. error bars (n = 3). (**B**) Nuclease time course of 2 nM Dna2 with 10 nM 5' RNA-DNA overhang or 5' DNA overhang substrates. ATP or AMPPNP is at 1.3 mM.

The following figure supplements are available for Figure 3:

**Figure supplement 1.** The ssDNA-length dependence of the DNA affinity correlates with cleavage rates.

(*Krajewski et al., 2014*; *Yeeles et al., 2009*). While most of these elements have diverged beyond a ~2 Å r.m.s.d., their secondary structures, arrangement and structural implications are closely related. In particular, both proteins have a β–α–α N-terminal extension and an α–α C-terminal extension that are stapled together by the iron-sulfur cluster (*Figure 2A*). This iron-sulfur cluster domain supports a loop that crosses over the catalytic channel and converts it to a tunnel through which the DNA has to thread. λ exonuclease and RecB also have a crossover loop, but their N- and C-terminal extensions that anchor it are structurally divergent from Dna2, and, more significantly, they lack the iron-sulfur cluster (*Singleton et al., 2004*; *Zhang et al., 2011*).

The Dna2 nuclease domain binds to a total of four phosphodiester groups, two before and two after the scissile phosphate group (*Figure 2B,C*). The scissile phosphate group contacts two calcium ions, one (Ca-1) through a non-bridging oxygen, and another (Ca-2) through the 3' bridging oxygen of the preceding base. The Ca-1 ion has an octahedral coordination shell very similar to other PD-(D/E)XK nucleases (*Pingoud et al., 2005*; *Zhang et al., 2011*). It's formed by the side chains of His164, Asp278 and Glu299, the Leu300 main chain carbonyl group, the non-bridging oxygen atom of the scissile phosphate group, and a water molecule, which also hydrogen bonds to Lys301 and is positioned for nucleophilic attack on the scissile phosphate (*Figure 2B* and *Figure 2—figure supplement 1A*). Lys301 is buttressed by Gln314, a motif IV residue characteristic of the RecB and λ exonuclease families (*Aravind et al., 2000*). The relative position of the second, Ca-2 ion differs from

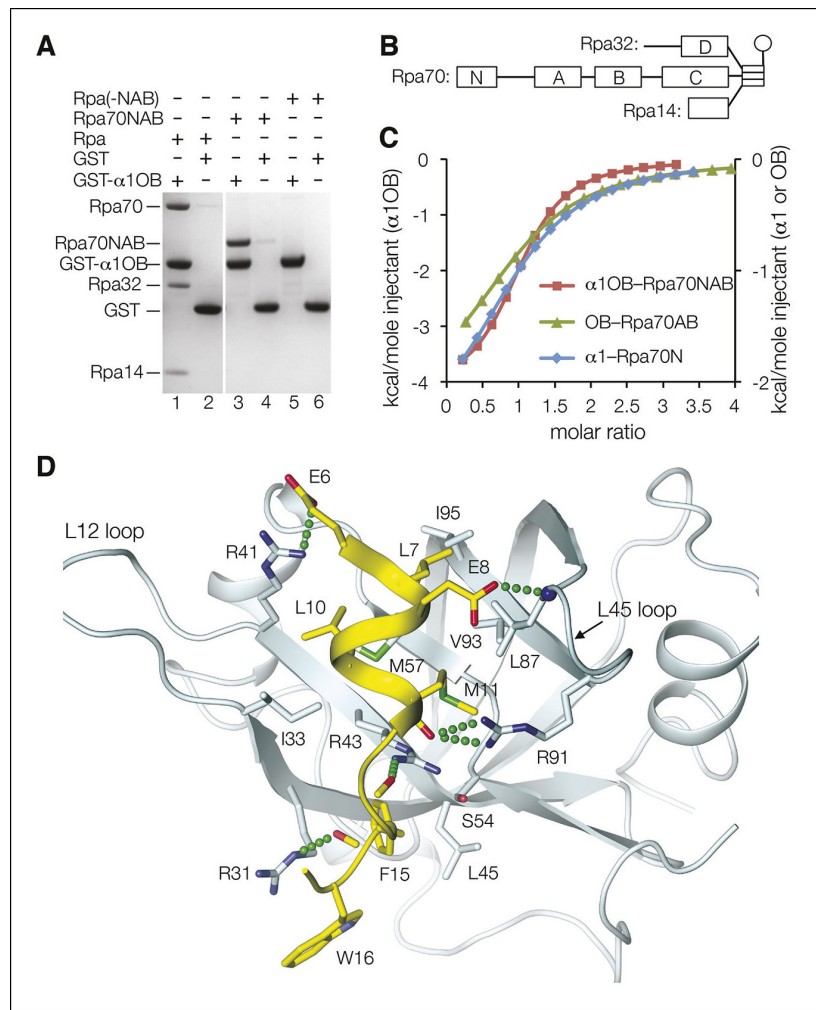

**Figure 4.** Dna2 physically interacts with Rpa. (**A**) GST pull-down assay showing that α1OB of Dna2 (residues 1–122) interacts with Rpa70NAB (1–431), but not the rest of Rpa heterotrimer (Rpa(-NAB)). (**B**) Schematic drawing of the Rpa trimer, showing the OB domains as rectangles and the winged helix (WH) domain of Rpa32 as a sphere. (**C**) ITC curves for the human α1OB-Rpa70NAB, α1-Rpa70N and OB-Rpa70AB complexes. (**D**) Structure of the α1-Rpa70N complex. The α1 peptide is in yellow and Rpa70N in cyan. For clarity, the main chain amide group of L87 and carbonyl groups of M11, K13, F15 are not labeled.

The following figure supplements are available for Figure 4:

**Figure supplement 1.** Comparison of the Dna2 α1–Rpa70 OBN, p53–Rpa70 OBN and Dna2 OB structures.

other nucleases (*Figure 2—figure supplement 1A*), and it is not clear whether it reflects the use of calcium instead of magnesium, or a divergent aspect of the nuclease mechanism of Dna2.

The base groups of the 6 nt segment stack in two sets of three, with Met157 and other crossover loop side chains wedging in between the Ade12 and Gua13 base groups (*Figure 2B*). This is very similar to the λ exonuclease crossover loop, which marks the transition from double stranded to single stranded and is thought to play a key role in unwinding dsDNA, although λ exonuclease does so without a helicase domain and in the context of a homo-trimeric assembly that contacts the dsDNA of the substrate (*Zhang et al., 2011*). Nevertheless, Met157 and the crossover loop may have an analogous function in the weak strand-separating activity of Dna2.

## Helicase structure

Dna2's combination of the β barrel, stalk and helicase domains, and their relative arrangement are strikingly similar to RNA/DNA helicases of the Upf1 subfamily, which contains Upf1, Ighmbp2 and Senataxin (*Fairman-Williams et al., 2010*). The ~600-residue assembly of the stalk, β barrel and helicase 1A and 2A domains can be superimposed on the structure of Ighmbp2 with a 1.5 Å Cα r.m.s.d. over 452 residues, with most of the non-superimposing residues accounted for by a ~10° rotation of the β barrel, which otherwise is structurally conserved (*Figure 2D*). The structural similarity between Dna2 and Ighmbp2 is actually more extensive than that between Ighmbp2 and Upf1 (1.8 Cα Å r.m.s. d for 436 residues) (*Chakrabarti et al., 2011*; *Lim et al., 2012*).

The Upf1-like subfamily is one branch of the SF1B family of helicases that translocate in the 5' to 3' direction (*Fairman-Williams et al., 2010*). The other SF1B branch, represented by the DNA helicase RecD2 (*Saikrishnan et al., 2009*), lacks the barrel and stalk domains. Furthermore, the individual RecD2 1A and 2A domains are not as similar to Dna2, with alignment r.m.s.d. values of 1.8 Å for 134 residues and 1.9 Å for 97 residues, respectively, in contrast to the Dna2-Ighmbp2 alignment, where the corresponding values are 1.5 Å for 204 residues and 1.4 Å for 183 residues, respectively. Together, these structural observations suggest that Dna2 evolved by incorporating an ancestral Upf1-family helicase.

DNA binds to Dna2 through both its phosphodiester and base groups (*Figure 2E*). Dna2-base interactions include Van der Waals contacts from the motif III loop (Leu795 and Val797), which wedges in between the bases of the last two 2A-bound nucleotides. These contacts are consistent with the proposed role of motif III in preventing DNA sliding during translocation of SF1B helicases (*Saikrishnan et al., 2009*). Dna2-phosphodiester interactions involve protein pockets that are rich in backbone amide and short-side chain hydroxyl groups (*Figure 2E*). Three consecutive phosphodiester groups, near the 5' end of the DNA, bind to the 2A domain (*Figure 2C,E*). The fourth phosphodiester group is in the cleft between the 2A and 1A domains and does not contact the protein, while the fifth and sixth phosphodiester groups bind to pockets on the 1A domain (*Figure 2C*). Contacts to the ribose groups are minimal, and they are consistent with the helicase accommodating the 2' hydroxyl group of RNA. In fact, the DNA contacts as well as the phosphodiester backbone conformation are very similar to those of the Ighmbp2-RNA, Upf1-RNA complexes (*Figure 2—figure supplement 1B,C*). This is in contrast to the RecD2-ssDNA structure, where extensive aromatic and van der Waals contacts with the sugar are thought to discriminate against RNA (*Saikrishnan et al., 2009*).

The helicase and ssDNA-dependent ATPase activities of Dna2 are considerably weaker than other helicases (*Budd et al., 2000*; *Masuda-Sasa et al., 2006*). One possible explanation, at least for the low ATPase/translocation rate, is the helicase domain being 5' to the nuclease domain on the DNA. This would, in principle, make completion of the ATPase/translocation cycle dependent on the nuclease domain releasing its grip on the DNA (see *Figure 2—figure supplement 1* legend for discussion of translocation).

## Mechanism of DNA binding

The structure explains why the nuclease activity of Dna2 requires the ssDNA to have a free end (*Bae and Seo, 2000*; *Kao et al., 2004*). The active site is embedded in a ~10 Å wide portion of the tunnel, and the tunnel entrances leading to it are too narrow to accommodate dsDNA or a single-stranded loop of a gap substrate (*Figure 1A,B*).

Since isolated Dna2 can degrade ssDNA with either 5' to 3' or 3' to 5' polarity in vitro, the ssDNA must be able to enter and thread through either end of the tunnel. In 5' flap processing, threading likely proceeds through initial, transient interactions at the nuclease domain entrance of the tunnel, with subsequent re-binding events occurring further along the tunnel interior. This threading is likely related to that of λ exonuclease, where a 5' terminal phosphate binding site inside the enzyme is proposed to drive the forward movement of the DNA in an electrostatic ratchet mechanism (*Zhang et al., 2011*).

The 5' to 3' nuclease activity was shown to require ~15 nts of ssDNA for optimal affinity and cleavage (*Bae et al., 2001*; *Gloor et al., 2012*). This suggests that the nuclease domain-DNA interactions do not provide sufficient binding energy or their half-life is too short relative to the catalytic step of cleavage. We thus presume that the 5' end of the ssDNA threads through the nuclease

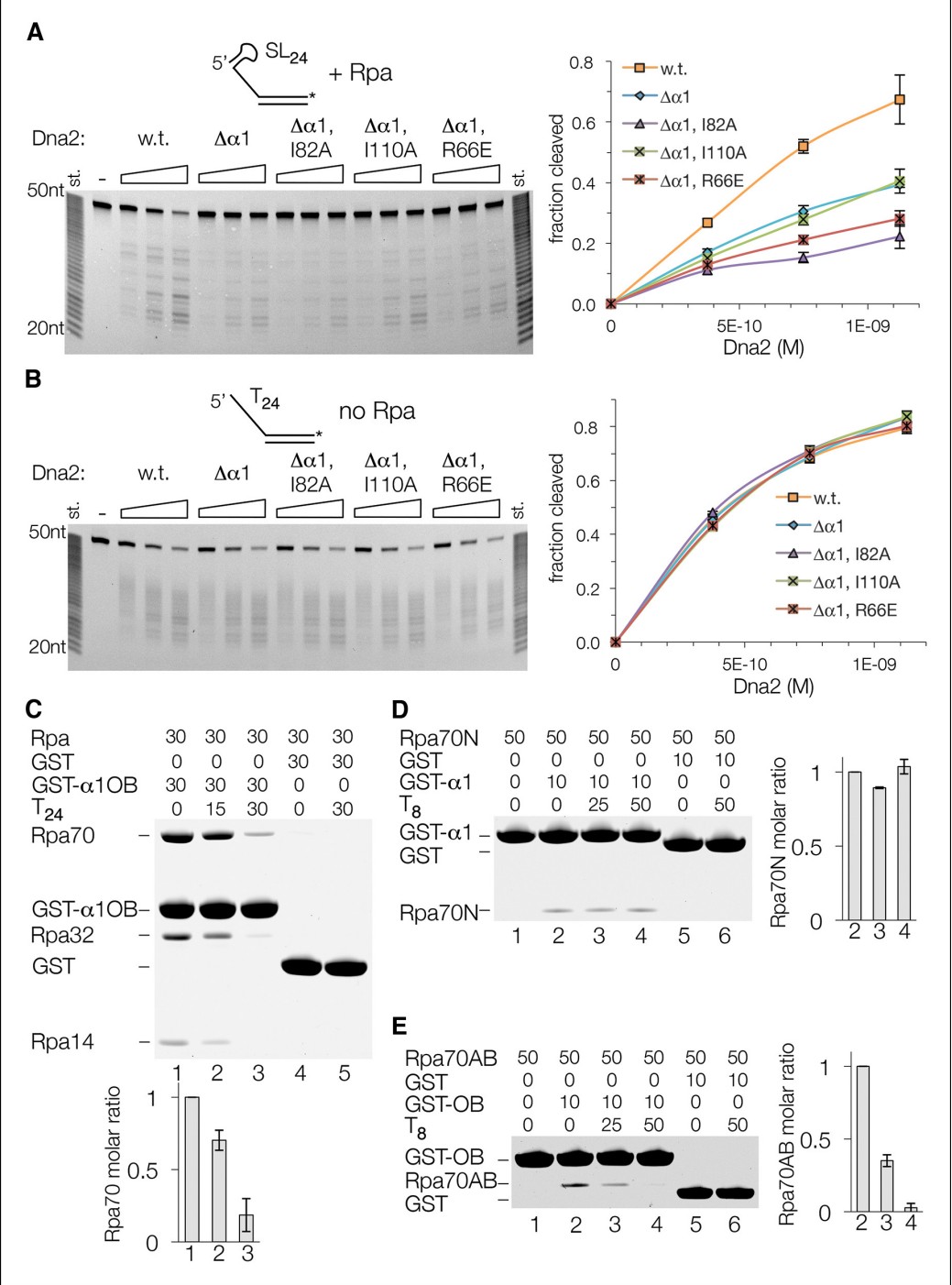

**Figure 5.** Both Dna2-Rpa interactions are important for Dna2 stimulation, but only one is mutually exclusive with Rpa-ssDNA interactions. (**A**) Cleavage of a 5' stem-loop overhang substrate (15 nM) by wild type, α1-deleted, and OB mutant Dna2 in the presence of 15 nM Rpa. (**B**) Nuclease activity of the same set of enzymes as in (**A**), but using a 5' $(dT)_{24}$ overhang substrate in the absence of Rpa. (**C**) GST pull-down assay showing that α1OB and $(dT)_{24}$ ssDNA bind to Rpa in a partially mutually exclusive manner. Protein and DNA concentrations are in μM, and the bar graph shows the quantitation of Rpa70 binding relative to the lane in the absence of DNA, which is set to 1. Error bars are standard deviations from three repetitions of each experiment. (**D**) The binding of α1 to the Rpa70N polypeptide is unaffected by $(dT)_8$ ssDNA. GST pull-down assay and quantitation as in (**C**). (**E**) The binding of OB to the Rpa70AB polypeptide is abolished by $(dT)_8$ ssDNA. GST pull-down assay and quantitation as in (**C**).

*Figure 5. continued on next page*

*Figure 5. Continued*

The following figure supplements are available for Figure 5:

**Figure supplement 1.** Cleavage of a ssDNA containing secondary structure is Rpa dependent.

---

portion of the tunnel without cleavage, until it reaches the helicase domain and engages the DNA-binding pockets there.

To investigate this in more detail, we assayed the Dna2 affinity and cleavage of a series of 5′ overhang substrates with ssDNA lengths that extend successively from the nuclease to the helicase 1A and 2A domains. The ssDNA consisted of deoxythymidine nucleotides, which have minimal secondary structure and thus do not require Rpa for cleavage by Dna2 ($5'(dT)_6$ to $5'(dT)_{24}$) (*Figure 3—figure supplement 1A*). The short substrates that can span only the nuclease domain ($5'(dT)_6$ and $5'(dT)_8$) have Dna2 affinities approximately two orders of magnitude lower than that of $5'(dT)_{24}$, and they exhibit minimal cleavage (*Figure 3—figure supplement 1A,B*). With the slightly longer $5'(dT)_8$, low level cleavage occurs 7 and 8 nucleotides from the 5′ end, indicating that the ssDNA extends to the helicase 1A domain (*Figure 3—figure supplement 1A*). The sites of cleavage indicate that Dna2 opens up ~2 base pairs (bps) of the duplex, since the tunnel entrance 3′ to the scissile phosphate is too narrow to accommodate double-stranded DNA. As ATP is neither required, nor has a significant effect, this DNA unwinding is the result of threading, likely analogous to DNA unwinding by λ exonuclease (*Zhang et al., 2011*). Extending the ssDNA by 2 nts, ($5'(dT)_{10}$), results in a major increase in Dna2 affinity and cleavage, with the cleavage sites indicating the engagement of the helicase 2A domain after the unwinding of ~2 bps (*Figure 3—figure supplement 1A*). With $5'(dT)_{17}$ where the ssDNA is long enough to reach both the 1A and 2A domains without duplex unwinding, there is a final increase in Dna2 affinity and cleavage that plateau at the levels of $5'(dT)_{24}$.

The 3′ to 5′ nuclease activity would require the 3′ end of the ssDNA to enter the tunnel at the helicase 2A domain. As the phosphodiester-binding sites of the 2A domain are fully accessible to bulk solvent, the structure suggests that 3′ end threading should be more efficient than 5′ end threading in the absence of ATP. Indeed, Dna2 cleaves a 3′ overhang $(dT)_{18}$ substrate at least 3-fold faster than the corresponding $5'(dT)_{18}$ (*Figure 3A*). The structure further suggests that the reported inhibition of 3′ end cleavage by ATP is due to 3′ end threading being counteracted by the helicase domain moving the DNA in the opposite direction (*Figure 3A*) (*Bae and Seo, 2000*; *Masuda-Sasa et al., 2006*).

Taken together, these findings support the model that a major role for the helicase domain is augmenting the DNA affinity of Dna2. As lack of ATP hydrolysis does not affect overall DNA binding by the helicase 1A and 2A domains, this model is consistent with the Dna2 ATPase-activity being dispensable for viability in yeast, and for DSB end resection in vitro, in contrast to the nuclease activity that is essential for both (*Bae and Seo, 2000*; *Budd et al., 2000*; *Cejka et al., 2010*; *Niu et al., 2010*; *Zhu et al., 2008*).

## Role of helicase activity in 5′ Okazaki fragment processing

RNA, which is not cleaved by Dna2, can substitute for part of DNA in the length-dependency of cleavage, suggesting that it can interact with at least part of Dna2 (*Bae et al., 2001*; *Bae and Seo, 2000*). In accord, we find that the affinity of Dna2 for a $5'(U)_{24}$ RNA overhang-DNA duplex is within an order of magnitude of its affinity for a comparable all-DNA substrate (*Figure 3—figure supplement 1B*). This observance, coupled with the structural similarity of Dna2 to the Upf1 subfamily of RNA/DNA helicases raises the possibility that the helicase activity facilitates the bypassing of the 5′ RNA primer of Okazaki fragments. Indeed, ATP but not the non-hydrolyzable ATP analogue AMPPNP stimulates the cleavage of a $5'(U)_{12}–(dT)_{12}$ RNA-DNA overhang substrate by ~50% (*Figure 3B*, top panel). By contrast, ATP had a minimal effect on the cleavage of the corresponding all-DNA $(dT)_{24}$ substrate, while AMPPNP inhibited cleavage slightly (*Figure 3B*, bottom panel).

## Dna2-Rpa association

Rpa is a heterotrimeric protein that consists of the Rpa70, Rpa32 and Rpa14 subunits. Genetic and biochemical studies in budding yeast indicated that Dna2 binds to the Rpa70 subunit, through an

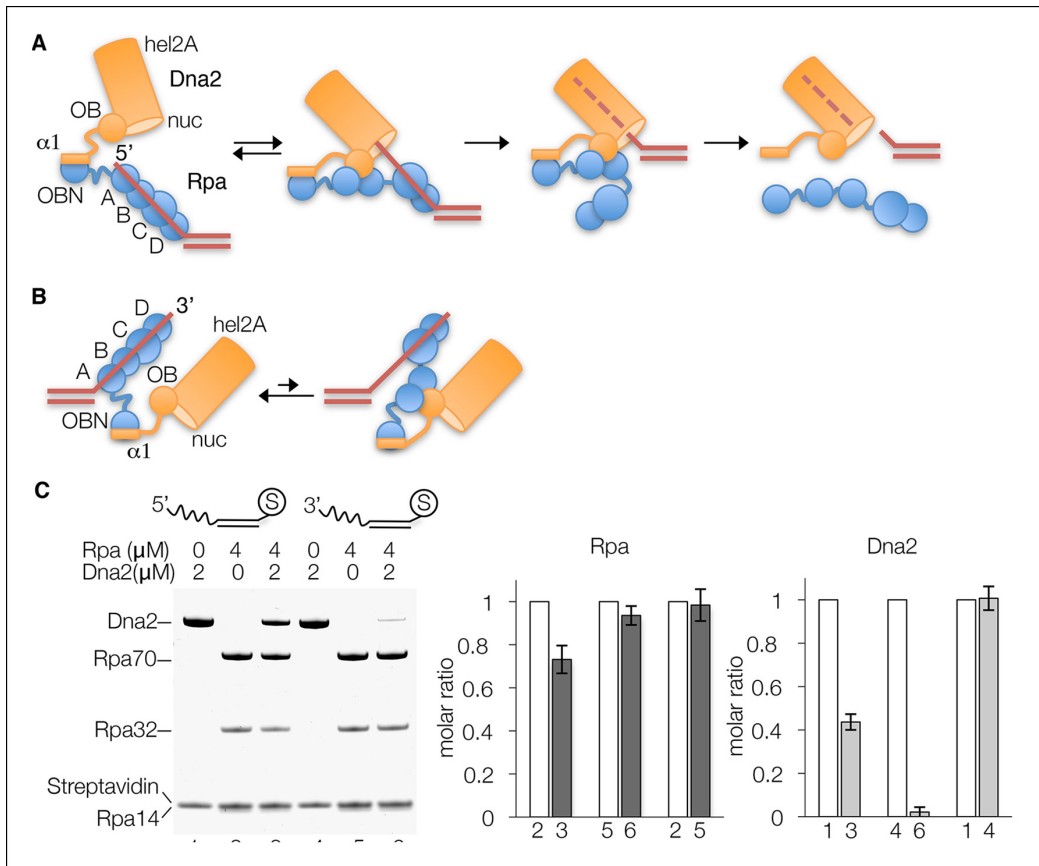

**Figure 6.** Dna2 displaces Rpa from 5' but not 3' overhang DNA. (**A**) Schematic of the proposed mechanism of Dna2 displacing Rpa from a 5' overhang DNA. Dna2 is shown as a hollow cylinder, except for its α1 helix and OB fold domains that are shown as a rectangle and circle, respectively. The label "nuc" marks the nuclease tunnel entrance into which the ssDNA would thread, and "hel2A" marks the helicase tunnel exit where the 5' end of the ssDNA would end up after threading. (**B**) Schematic illustrating that at a 3' overhang DNA, Dna2-Rpa interactions do not result in a free ssDNA end that can thread into the Dna2 tunnel. (**C**) Nuclease-dead Dna2 (D278A) displaces Rpa from 5'- but not 3' overhang DNA. The overhang consists of (dT)$_{26}$ and the DNA is conjugated to streptavidin (shown by "S") through a biotin group at the end of the duplex. Column graph showing quantitation of Rpa32 plots the molar ratio relative to the reaction lacking Dna2 for each DNA substrate (lanes 2, 3 for 5' overhang DNA, and lanes 5 and 6 for 3' overhang DNA), or relative to lane 1 for the comparison of Rpa loading onto 5'- and 3'-overhang DNA. Quantitation of relative Dna2 loading is similarly shown in the last column graph. Error bars are standard deviations from three repetitions of each experiment.

interaction between the N-terminal portions of the two proteins (*Bae et al., 2003*). These studies further pointed to additional binding sites on Dna2 and Rpa70, as deletion of the N-terminal interacting segments reduced but did not eliminate Dna2-Rpa association and the stimulation of the nuclease activity (*Bae et al., 2003*). The N-terminal portion of yeast Dna2 (residues 1 to 405) is poorly conserved in mammalian orthologs and also contains a ~350-residue yeast-specific extension, although it appears to contain an acidic/amphipathic helix analogous to α1 of mouse Dna2, and likely encompasses a portion of the OB domain (*Figure 1—figure supplement 1A*).

We thus tested whether a mouse Dna2 fragment consisting of the α1 helix and OB domain (thereafter α1OB; residues 1 to 122) binds to mouse Rpa using a GST pull-down assay. As shown in *Figure 4A*, GST-α1OB but not isolated GST binds to heterotrimeric Rpa in a 30 μM stoichiometric solution (lanes 1 and 2).

The Dna2-interacting N-terminal segment of yeast Rpa70 (residues 1 to 180) consists of an OB fold domain (named OBN) that is a known protein-protein interaction site and a ~60 residue flexible linker (*Figure 4B*) (*Bochkareva et al., 2005*; *Fan and Pavletich, 2012*; *Fanning et al., 2006*;

*Gomes and Wold, 1996*). The rest of Rpa70 consists of three OB folds that are the main DNA-binding domains (named DBD-A, DBD-B and DBD-C; *Figure 4B*) (*Bochkarev et al., 1997*; *Fan and Pavletich, 2012*). Because DBD-A and DBD-B can also participate in protein-protein interactions (*Jiang et al., 2006*; *Loo and Melendy, 2004*; *Yuzhakov et al., 1999*), this raised the possibility that they account for the remainder of Rpa70's Dna2 affinity.

We tested this using a mouse Rpa70 fragment containing the OBN, DBD-A and DBD-B domains (residues 1 to 431; thereafter Rpa70NAB). As shown in *Figure 4A*, GST-α1OB binds to Rpa70NAB (lanes 3 and 4) but not to the Rpa heterotrimer lacking this fragment (Rpa(-NAB); lanes 5 to 6). The dissociation constant ($K_d$) of the α1OB–Rpa70NAB complex, determined by isothermal titration calorimetry (ITC), is 12 ± 1 μM (*Figure 4C* and *Figure 4—figure supplement 1E*). Using ITC, we found that the Dna2 α1 helix (residues 1 to 20) binds to the OBN domain of Rpa70 (residues 1 to 120; thereafter Rpa70N), while the Dna2 OB domain (residues 21 to 122) binds to the Rpa70 fragment containing DBD-A and DBD-B (residues 181 to 422; thereafter Rpa70AB), with $K_d$ values of 34 ± 7 and 46 ± 10 μM, respectively (*Figure 4C* and *Figure 4—figure supplement 1*). The $K_d$ values of these subcomplexes relative to that of α1OB–Rpa70NAB indicate only a low level of cooperativity, consistent with the two interacting elements on both Rpa and Dna2 being separated by flexible linkers. Further dividing Rpa70AB into the individual DBD-A and DBD-B polypeptides failed to show detectable binding to Dna2 OB under the same conditions, indicating both are required (not shown).

## Structure of the Dna2 α1 helix bound to the OBN domain of Rpa70

We co-crystallized a human Dna2 peptide (residues 1 to 20) containing the α1 helix with the OBN domain of human Rpa70 (residues 1 to 120). In the 1.6 Å structure, residues 6 to 17 of Dna2 form a 2-turn amphipathic helix followed by a β turn, while the rest are disordered (*Figure 4D*). The peptide binds to a shallow OBN groove that corresponds to the DNA-binding grooves of other OB domains (*Bochkarev et al., 1997*; *Fan and Pavletich, 2012*). The only other OBN-peptide structure available is of OBN fused to the p53 transactivation domain peptide, where two p53 peptides occupy the OBN groove (*Bochkareva et al., 2005*). The amphipathic helix of Dna2 overlaps with one of the p53 peptides, while the β turn coincides with part of the other p53 peptide (*Figure 4—figure supplement 1A*).

The mixed basic and hydrophobic character of the OBN groove complements the acidic-hydrophobic nature of the Dna2 peptide. Four arginine guanidinium groups and one backbone amide group of OBN contact two side chain carboxylate and three backbone carbonyl groups from α1 (*Figure 4D*). One of these OBN arginine residues (Arg43) splits the otherwise hydrophobic groove, demarcating two hydrophobic pockets. One pocket (Ile33, Met57, Leu87, Val93 and Ile95) binds to the hydrophobic face of the Dna2 helix (Leu7, Leu10 and Met11), while the other pocket (Leu45 and aliphatic portions of Arg31, Arg43, and Ser54) binds to Phe15 and Trp16 from the β turn of Dna2.

We do not know the structure of the Dna2 OB domain bound to the Rpa70 DBD-A and DBD-B domains, but the OB structure in intact Dna2 is consistent with a role in protein-protein interactions (*Figure 4—figure supplement 1B*), and the isolated OB domain polypeptide does not exhibit any DNA-binding in EMSA at concentrations up to 80 μM (*Figure 4—figure supplement 1F*).

## Mechanism of Rpa displacement by Dna2

A common feature of proteins involved in Rpa-dependent processes is their ability to bind to Rpa, either directly or through accessory factors, and this is thought to form the basis for the displacement of Rpa from ssDNA (*Fanning et al., 2006*; *Zou et al., 2006*). Rpa displacement is best understood with the simian virus 40 (SV40) T antigen and related viral replication proteins, where T antigen-Rpa interactions allosterically modulate Rpa's ssDNA affinity (*Jiang et al., 2006*; *Loo and Melendy, 2004*; *Yuzhakov et al., 1999*). Because Dna2 has been shown to displace Rpa from 5' flap DNA (*Stewart et al., 2008*), we sought to address whether Dna2-Rpa interactions have an analogous, direct role in Rpa displacement, or whether they reflect a simple recruitment process that allows Dna2 to better compete with Rpa for ssDNA.

We first confirmed that both the α1 helix and OB domain of Dna2 are required for the stimulation of the nuclease activity by Rpa. For this, we used a 5' overhang substrate with a stem loop secondary structure that makes the Dna2 nuclease activity dependent on Rpa (5'SL$_{24}$) (*Figure 5—figure*

supplement 1A). In keeping with the findings with the yeast Dna2△405N mutant, deletion of the Dna2 α1 helix reduced cleavage of the Rpa-coated 5'SL$_{24}$ by a factor of ~2 compared to intact Dna2 (*Figure 5A*). The analogous experiment to address the importance of the OB domain was not possible, as the OB-deleted Dna2 is insoluble (not shown). We instead mutated three OB residues at positions corresponding to protein-protein contacts on the Rpa70 OBN domain (*Figure 4—figure supplement 1B*). As shown in *Figure 5A*, two of these mutations (I82A and R66E) synergized with α1 deletion and reduced 5'SL$_{24}$ cleavage further, whereas the third (I110A) had no discernible effect. To rule out that these mutations do not affect the structural integrity of Dna2, we tested the Rpa-independent cleavage of the 5'(dT)$_{24}$ substrate and found that it is not affected by neither the OB mutations nor the α1 deletion (*Figure 5B*).

We then addressed whether the α1 helix and OB domain play a direct role in Rpa displacement, which in principle can occur either through the allosteric destabilization of Rpa-ssDNA interactions as shown for T antigen, or through α1OB binding to an Rpa site that overlaps with or sterically hinders a ssDNA-binding site. We did not expect the isolated α1OB, in the absence of the DNA-affinity provided by the nuclease and helicase domains, to displace Rpa from ssDNA, as the affinities of the α1OB-Rpa and Rpa-ssDNA complexes differ by 5 orders of magnitude ($K_d$ values of ~12 µM and ~100 pM, respectively). Instead, we reasoned that if α1OB has a role destabilizing Rpa-ssDNA interactions, then this should be reflected in ssDNA interfering with α1OB-Rpa association. As shown in *Figure 5C*, addition of (dT)$_{24}$ substantially reduced Rpa binding to GST-α1OB (lanes 1 to 3), consistent with α1OB and ssDNA interacting with Rpa in a mutually exclusive manner. However, while one molar equivalent of (dT)$_{24}$ reduced the bound Rpa by a factor of ~4, the remaining Rpa was clearly above the background level of the GST-only reaction (lanes 4 and 5), suggesting that only one of the two Dna2-Rpa interactions is mutually exclusive with Rpa-ssDNA interactions. Consistent with this, addition of (dT)$_8$ had no discernible effect on the binding of GST-α1 to Rpa70N (*Figure 5D*), whereas it eliminated the binding of GST-OB to Rpa70AB in a manner dependent on the stoichiometry of (dT)$_8$ to Rpa70AB (*Figure 5E*).

These findings indicate that the interaction between the Dna2 α1 helix and the Rpa OBN domain is a simple recruitment step, consistent with both of these elements being flexibly tethered to the remainder of their polypeptides and with their lack of ssDNA affinity. This simple recruitment would be important for Dna2 accessing ssDNA-bound Rpa, where the interaction between the Dna2 OB and Rpa DBD-A–DBD-B domains would not be initially available. The simple recruitment interaction would also increase the effective concentration of the Dna2 OB at the Rpa DBD-A–DBD-B, as they immediately follow the α1 helix and OBN domain, respectively (*Figure 6A*).

This juxtaposition would then increase the probability of DBD-A–DBD-B transiently associating with OB and dissociating from ssDNA. This is plausible, because while the DNA affinity of DBD-A–DBD-B (~50 nM $K_d$) is substantially higher than its Dna2 affinity, those of the individual DBD-A (2 µM $K_d$) and DBD-B (20 µM $K_d$) are not, and as with the intact Rpa heterotrimer, they are thought to associate with and dissociate from DNA sequentially (*Arunkumar et al., 2005*; *Fan and Pavletich, 2012*; *Fanning et al., 2006*). By itself, the transient displacement of DNA from DBD-A–DBD-B will not lead to the release Rpa from ssDNA. However, because the DBDA–DBDB is at the 5' end of the Rpa-DNA complex (*Bochkarev et al., 1997*; *Fan and Pavletich, 2012*), and the Dna2 OB domain is next to the nuclease tunnel entrance, the transiently free 5' end of the DNA will be well-placed to start threading through the nuclease tunnel (*Figure 6A*). The threading process then should be able to completely dissociate the already weakened Rpa-DNA complex (*Figure 6A*). At a 3' end of DNA, by contrast, the transient dissociation of DBD-A–DBD-B from DNA will expose an internal ssDNA lacking an end that can be trapped by the Dna2 tunnel, and DBD-A–DBD-B will revert to their DNA-bound state (*Figure 6B*).

This mechanism of Rpa displacement predicts that the inhibition of 3' end cleavage by Rpa is due to the inability of Dna2 to displace Rpa there. To test this prediction, we conjugated 5'- or 3'-(dT)$_{26}$ overhang DNA that was biotinylated on the duplex end onto streptavidin beads, loaded it with Rpa, and then added nuclease-dead Dna2 (D278A mutant). As shown in *Figure 6C*, Dna2 reduced the amount of Rpa bound to the 5' overhang DNA by ~30% (lanes 2 and 3), whereas it had a minimal effect on the Rpa bound to the 3' overhang DNA (~5% reduction; lanes 5 and 6). The amount of Rpa displacement was proportional to Dna2 loading, which was substantial with 5' overhang DNA (~45% of the reaction lacking Rpa; lanes 1 and 3), but negligible with 3' overhang DNA (~3%; lanes 4 and 6). In the absence of Rpa, by contrast, the amount of Dna2 loading on the two DNA substrates was

essentially identical (lanes 1 and 4), consistent with the inability of Dna2 to load onto the 3' overhang being due to its failure to displace Rpa.

## Conclusions

The Dna2-ssDNA structure shows that the active site and most of the DNA-binding sites are enclosed in a narrow tunnel, necessitating the threading of the DNA through a tunnel end to access the DNA binding sites. The structure also indicates that the translocase activity of the helicase domain does not drive threading. A 5' DNA end would have to thread halfway through the tunnel before it can access the helicase domain, while the threading of a 3' end starting at the helicase 2A domain would be opposed by the 5' to 3' polarity of translocation, as we demonstrate. Instead of translocation, the helicase domain appears to be important for providing DNA affinity and for bypassing the 5' RNA primer of Okazaki fragments.

The structure also precludes the helicase domain tracking on DNA to any significant extent, because the nuclease domain is ahead, in the 5' to 3' direction of translocation. This is consistent with in vitro studies showing the ATPase activity to be dispensable for 5' flap processing and DSB resection (*Cejka et al., 2010*; *Niu et al., 2010*; *Zhu et al., 2008*), and the fact that this activity is rather low compared to bona-fide helicases (*Bae and Seo, 2000*; *Masuda-Sasa et al., 2006*). In yeast, ATPase mutations do result in growth defects (*Budd et al., 2000*), and it is possible this is due to the ATPase activity contributing to the bypassing of the RNA primer of Okazaki flaps, as sug-gested by our in vitro data. This may be reflected in the similarity of the Dna2 helicase domain to the Upf1 family of RNA/DNA helicases, which is extensive enough to indicate that Dna2 picked up an ancestral Upf1-like helicase during its evolution.

The requirement for threading necessitates Dna2 having a mechanism to displace Rpa from ssDNA. As our proposed mechanism of Rpa displacement predicts, we find that Dna2 can displace Rpa from a 5' but not a 3' end, explaining how Rpa dictates the proper end polarity of the nuclease activity of Dna2.

## Materials and methods

### Protein expression and purification

Full-length mouse Dna2 was cloned into a pFastbac1 baculovirus vector engineered with a cleavable N-terminal GST tag and a non-cleavable C-terminal FLAG tag, and was expressed in Hi-5 insect cells (Life technologies, Carlsbad, CA). The recombinant protein was purified first by GST-affinity chroma-tography and, after cleavage of the GST tag, by anion exchange and gel-filtration chromatography. Purified Dna2 was concentrated to ~20 mg/mL in 20 mM Tris-HCl, 250 mM NaCl, 0.3 mM TCEP, pH 8.0. All buffers were degassed before use. The various Dna2 mutants and seleno-methionine substi-tuted protein were purified similarly. Seleno-methionine substituted Dna2 was expressed according to manufacturer's protocol (Expression systems, Davis, CA) and was purified similarly.

For the expression of the mouse Rpa heterotrimer, Rpa70 was cloned into a pFastbac1 vector and Rpa32/Rpa14 were cloned into a modified pFastBac-dual vector with Rpa32 fused to a cleavable N-terminal GST-tag. The Rpa heterotrimer was produced by co-infecting Hi-5 cells with both viruses, and was purified as described (*Fan and Pavletich, 2012*). The Rpa heterotrimer with truncated Rpa70 was expressed and purified similarly.

GST-tagged mouse Dna2 $\alpha$1 (residues 1–20), $\alpha$1OB (residues 1–122) and OB (residues 21–122) and Rpa70NAB (residues 1–431), Rpa70N (residues 1–122), and Rpa70AB (residues 191–431) frag-ments, as well as the corresponding human polypeptides used in ITC measurements, were cloned into a pGEX-4T vector and expressed in *E. coli* BL21DE3 cells. They were purified by GST affinity chromatography, ion exchange and/or heparin chromatography, and gel-filtration chromatography. The corresponding untagged polypeptides were expressed fused to an N-terminal 6-His-sumo tag in *E. coli* BL21DE3 cells. Following nickel affinity chromatography and cleavage of the 6-His-sumo tag by Ulp1, they were further purified by ion exchange and/or heparin, and gel-filtration chromatography.

## Crystallization

Crystals of the Dna2-ADP complex were grown in 4°C using the hanging drop vapor diffusion method from a crystallization buffer of 80 mM MES, 250 mM $Li_2SO_4$, 2 mM $MgCl_2$, 8–12% PEG MME 5000, 0.5 mM TCEP, pH 6.5, containing 12 mg/mL Dna2 and 1 mM ADP. Seleno-methionine substituted Dna2 was crystallized under similar conditions using seeding. Crystals of Dna2 bound to 21-nt ssDNA and ADP (Dna2-ssDNA in *Table 1*) were grown from a crystallization buffer of 80 mM MES, 20 mM $CaCl_2$, 10 mM spermidine, 4–9% isopropanol, 0.5 mM TCEP, pH 6.5, and 1 mM ADP. They contain two molecules in the asymmetric unit. Crystals of Dna2 bound to 5′ overhang DNA, which consists of a 17-nt 5′ overhanging ssDNA and a 6 base pair dsDNA (Dna2-5′ overhang DNA in *Table 1*), grew from a similar condition but in a different space group, and have one complex in the asymmetric unit and higher diffraction limits. As there is no electron density for the duplex, we presume it is disordered. All crystals were cryo-protected in crystallization buffer supplemented with 20–25% glycerol or ethylene glycol, and were flash-frozen in liquid nitrogen. The human DNA2 α1-RPA70 OBN complex was crystallized by mixing an 8.7 mg/ml solution of the RPA70N polypeptide (residues 1–120) with a 3-fold molar excess of a synthetic DNA2 α1 peptide (residues 1–20) from 50 mM Tris-HCl, 35% PEG 1500, 2 mM TCEP, pH 8.0. Crystals were cryo-protected in crystallization buffer supplemented with 20–25% glycerol and flash-frozen in liquid nitrogen.

## Structure determination and refinement

Diffraction data were collected at the 24IDC and 24IDE beamlines of the Advanced Photon Source (Argonne National Laboratory) and the X29 beamline of the National Synchrotron Light Source (Brookhaven National Laboratory). Data sets were processed with the HKL2000 suite (*Otwinowski and Minor, 1997*). The structure of Dna2-ADP complex was determined using SAD with data collected at the selenium edge (*Bricogne et al., 2003*). The phases were improved using solvent flattening and two-fold NCS averaging with multiple masks with the program DM (*Winn et al., 2011*). The model was built using O (*Jones et al., 1991*)and Coot (*Emsley et al., 2010*) and refined with REFMAC5 (*Winn et al., 2011*) and PHENIX (*Adams et al., 2010*) using tight NCS restraints on atom positions. Initial phases for the two Dna2-ADP-ssDNA complexes were obtained by molecular replacement with PHASER (*McCoy et al., 2007*) using the apo-Dna2 structure as the search model, and the structures were refined using REFMAC5 (*Winn et al., 2011*) and PHENIX (*Adams et al., 2010*), with TLS parameterization of temperature factors of the high resolution Dna2-5′ overhang DNA complex. The Ramachandran plot of the final model has 90.5%, 8.9%, 0.5% and 0% of the residues in the most favored, additional allowed, generously allowed and disallowed regions. The statistics from data collection and refinement are shown in *Table 1*. Figures were generated using PyMOL (http://www.pymol.org).

## Nuclease assays

Unless otherwise noted, experiments were performed in a 15 µL volume in 20 mM Tris-HCl, 125 mM NaCl, 6 mM $MgCl_2$, 1.3 mM ATP, 0.2 mg/mL BSA, 2% glycerol, pH 8.0. Reactions were incubated for 30 min at 25°C and stopped by adding 0.5% SDS, 20 mM EDTA and 1 unit of proteinase K. Reactions were analyzed by 16% or 12% denaturing urea-PAGE. For Rpa-containing reactions, Rpa was incubated with DNA for 15 min at 4°C before the addition of Dna2. For reactions using 6-FAM labeled DNA, wet gels were scanned using a fluorescent laser scanner (Fujifilm FLA 5000), and the bands were quantified with ImageGauge software (Fujifilm).

## Protein-protein interaction assays

For GST pull-down experiments, 30 µM GST-tagged mouse α1-OB (residues 1 to 122) or GST was incubated with an equal molar amount of full-length Rpa, Rpa70NAB or Rpa(-NAB). Binding reactions (40 µL) were carried out in 20 mM Tris-HCl, 80 mM NaCl, 0.3 mM TCEP, 2% glycerol, pH 8.0 at 4°C for 30 min before addition of glutathione beads. After 30 min, the beads were washed three times with binding buffer. Proteins were eluted with 20 mM glutathione and analyzed by SDS-PAGE. Other GST pull-down experiments were carried out similarly, with protein concentrations indicated in the main text or figure legends.

For Rpa displacement experiments, 2 µM 5′- or 3′-$(dT)_{26}$ overhang DNA that was biotinylated at the duplex end was coupled to magnetic streptavidin beads. After washing, the beads were

incubated with a 1 molar equivalent of Rpa for 20 min in 20 mM Tris-HCl, 125 mM NaCl, 2% glycerol, 0.3 mM TCEP, 0.01% Tween 20, pH 8.0, followed by the addition of a one molar equivalent of Rpa or Rpa-Dna2 mixture. The beads were incubated for 45 min with mixing, and after 3 washes the beads were boiled and analyzed by SDS-page.

ITC experiments were carried out using a MicroCal ITC200 calorimeter (Malvern Instruments Inc., Westborough, MA) at 20˚C in a buffer of 20 mM HEPES, 80 mM NaCl, 0.2 mM TCEP, pH 7.5.

## Electrophoretic mobility shift assay

Binding reactions (10 μl) containing 0.075 or 0.3 nM of $^{32}$P-labelled DNA substrates with increasing amounts of nuclease-dead Dna2 were carried out in 20 mM Tris-HCl, 80 mM NaCl, 0.3 mM TCEP, 0.2 mg/mL BSA (New England Biolabs, Ipswich, MA), 2% glycerol, pH 8.0. Reactions were incubated on ice for 30 min., followed by electrophoresis at 4˚C on 4% (w/v) native PAGE gels in 1x TB buffer. The dried gels were scanned using a phosphorimager (GE Typhoon 7000, GE Healthcare, Pittsburg, PA), bands were quantified with ImageGauge software (Fujifilm), and the apparent dissociation constants ($K_d$) were calculated from the equilibrium expression of a one-site binding model. Curve fitting was done by minimizing the sum of the square of the differences between the observed fraction of bound DNA and the fraction predicted from the model.

## Acknowledgements

We thank Dr. David King and Dr. Hediye Erdjument-Bromage for mass spectroscopic analysis and the staff at the Advanced Photon Source NE-CAT (24IDC, E) and Brookhaven National Laboratory (X29) for beamline support. This work was supported by the Howard Hughes Medical Institute.

## Additional information

### Funding

| Funder | Author |
| --- | --- |
| Howard Hughes Medical Institute | Nikola P Pavletich |

The funders had no role in study design, data collection and interpretation, or the decision to submit the work for publication.

### Author contributions

CZ, SP, Conception and design, Acquisition of data, Analysis and interpretation of data, Drafting or revising the article; NPP, Conception and design, Analysis and interpretation of data, Drafting or revising the article

## Additional files

### Major datasets

The following datasets were generated:

| Author(s) | Year | Dataset title | Dataset ID and/or URL | Database, license, and accessibility information |
| --- | --- | --- | --- | --- |
| Zhou C, Pourmal S, Pavletich NP | 2015 | Crystal structure of Dna2 in complex with a 5' overhang DNA | http://www.rcsb.org/pdb/explore/explore.do?structureId=5EAN | Publicly available at the Protein Data Bank (accession no. 5EAN) |
| Zhou C, Pourmal S, Nikola P Pavletich | 2015 | Crystal structure of Dna2 in complex with an ssDNA | http://www.rcsb.org/pdb/explore/explore.do?structureId=5EAX | Publicly available at the Protein Data Bank (accession no. 5EAX) |

| Zhou C, Pourmal S, Pavletich NP | 2015 | Crystal structure of Dna2 nuclease-helicase | http://www.rcsb.org/pdb/explore/explore.do?structureId=5EAW | Publicly available at the Protein Data Bank (accession no. 5EAW) |
|---|---|---|---|---|
| Zhou C, Pourmal S, Pavletich NP | 2015 | Crystal structure of a Dna2 peptide in complex with Rpa 70N | http://www.rcsb.org/pdb/explore/explore.do?structureId=5EAY | Publicly available at the Protein Data Bank (accession no. 5EAY) |

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
