## [Decision Letter]

Thank you for submitting your work entitled "Dna2 nuclease-helicase structure, mechanism and regulation by Rpa" for peer review at *eLife*. Your submission has been favorably evaluated by John Kuriyan (Senior Editor) and two reviewers, one of whom, James Berger, has agreed to reveal his identity.

The editor and the reviewers have discussed the reviews with one another and the editor has drafted this decision to help you prepare a revised submission.

This manuscript from the Pavletich laboratory on Dna2 is a remarkable tour de force. The paper describes a structure/function analysis of the Dna2 ATPase/nuclease and its interaction with RPA. It is well-written and fun to read. The structure has multiple interesting features. The paper establishes how the PD-(D/E)XK nuclease and SF1B helicase domains of Dna2 jointly bind ssDNA and how Dna2 interacts with RPA physically and biochemically to process DNA ends. The results furthermore provide the structural mechanisms for how Dna2 works on both 3' and 5' overhangs. The additional insight from biochemical experiments on how Dna2 might displace RPA from 5' but not 3' overhangs is a bonus. This work will be of tremendous and broad-based biological interest across many fields. It is a great paper in multiple respects. *eLife* is a highly suitable venue for publication.

The reviewers have raised the following points. The authors should use their judgement in how best to address these issues in preparing a revised version of the paper.

1) Dna2 is a multi-domain protein. TLS may improve the refinement. Given the importance of this structure, optimizing the refinement is worth the extra effort.

2) Is there any structural feature that sets the register of the ssDNA? It is remarkable that there weren't mixed populations, one or two nt off. A comment on this would be useful.

3) The authors pointed out that Dna2 primarily contacts the phosphodiester portion of the DNA (paragraph three, subheading “Helicase structure”). However, it is intriguing that the nuclease seems to be phosphodiester dominated, while the helicase domain has contacts with the bases as well. This is likely functionally relevant. Could the authors add a comment?

4) Why wasn't there a directly comparable DNA substrate to the RNA (21 nt) in subheading “Role of helicase activity in 5’ Okazaki fragment processing”? The authors used a 24 nt substrate. For a non-specific DNA binding protein like Dna2, a few nt can make a difference.

5) As Dna2 must cooperate on flaps with FEN1, are there any useful insights from this structure as to how this coordination may work? Also how do the active site metal ions and bound ssDNA compare to the FEN1-DNA structure as both enzymes incise 5' flaps (Cell 2011 145(2): 198–211). What is similar and what is different in terms of the DNA binding and catalytic site? In considering the lower activity with short 5' flaps, one wonders why is it active? The structure shows how Fen1 is controlled. Could Dna2 be "activated" somehow? Also FEN1 has significant changes upon DNA binding. For Dna2, the overall Rmsd was 0.9, but do some side chains in the nuclease have significant shifts with and without DNA?

6) Given the lowered incision activity with small ssDNA oligonucleotides, can the authors explain the 5 nt limit on 5' flaps? Is it simply that once bound, Dna2 can easily incise down to 5 nt in a processive fashion? Is it possible to distinguish the processive incision rate from an initial bind-incise rate?

7) The displacement of RPA from the 5' end by Dna2 is interesting. However, the mechanism is difficult to understand, as the alpha1 and the OB fold are closer to where the 5' end ends up after threading than where the 5' end must thread in. Are there any added insights that can be gained from adding in the DNA-bound RPA structure, with a linker added in between 70N.

8) The models put forth for Dna2 action often invoke a need to thread DNA (or RNA) through the nuclease tunnel. The entirety of the tunnel would seem rather long and narrow for this to readily occur (e.g., in RecBCD/AddAB systems, movement toward the nuclease through a tunnel is aided by the helicase domains). Is it possible that the barrel and stalk might transiently undock from the body of Dna2 to make binding more favorable to the SF1B domain, and this binding could then aid threading through the (now shorter) tunnel? Also, such a conformational change occurs in the FEN1 structure where flexibility evidently aids threading prior to the formation of the active complex, so perhaps this merits a comparative comment.

9) Along similar lines, although it is true that the ATPase activity of Dna2 is not strictly essential for yeast viability, it is unclear from the model depicted in Figure 6 why Dna2 should need an ATPase domain at all. Given the conservation of this region in Dna2 homologs, this question merits some discussion/clarification.

10) Paragraph five, subheading “Overall structure of the Dna2-ssDNA complex”: Where does the α1 helix sit on the structure when it docks against a symmetry mate? Is this internal interaction functionally important? Would it be possible for this interaction to occur in cis?

11) Ramachandran values are not included in the Methods or in Table 1.

---

## [Author Response]

1) Dna2 is a multi-domain protein. TLS may improve the refinement. Given the importance of this structure, optimizing the refinement is worth the extra effort.

We now report the statistics from the TLS refinement, which has only a minor effect on the statistics and quality of electron density, consistent with the multiple domains being rigidly coupled.

*2) Is there any structural feature that sets the register of the ssDNA? It is remarkable that there weren't mixed populations, one or two nt off. A comment on this would be useful.*

We think the DNA binds in a single register because of base-base stacking effects. We now mention this possibility in the revised manuscript:

“The DNA bases stack continuously, except for a base step at the helicase, one at the nuclease and one at the transition between the two domains (Figure 2). Two of the three unstacked base steps are at pyrimidine-pyrimidine pairs, and this may contribute to the DNA binding at a well-defined register.” (paragraph four, subheading “Overall structure of the Dna2-ssDNA complex”).

We disfavor the alternative possibility of end effects, as unstructured or poorly-defined nucleotides exist at both the 5’ (one) and 3’ (three) ends of the DNA, and we now mention this in the revised manuscript (subheading “Overall structure of the Dna2-ssDNA complex”). *3) The authors pointed out that Dna2 primarily contacts the phosphodiester portion of the DNA (paragraph three, subheading “Helicase structure”). However, it is intriguing that the nuclease seems to be phosphodiester dominated, while the helicase domain has contacts with the bases as well. This is likely functionally relevant. Could the authors add a comment?*

We thank the reviewers for pointing out that our previous discussion of the functional importance of helicase-base contacts could be overlooked (“Van der Waals contacts include those […] during translocation of SF1B helicases”; subsection “Helicase structure”).

To fix this, we have moved the aforementioned sentence closer to the beginning of the paragraph and have modified the introductory sentence:

“DNA binds to Dna2 through both its phosphodiester and base groups […] that are rich in backbone amide and short-side chain hydroxyl groups…” *4) Why wasn't there a directly comparable DNA substrate to the RNA (21 nt) in subheading “Role of helicase activity in 5’ Okazaki fragment processing”? The authors used a 24 nt substrate. For a non-specific DNA binding protein like Dna2, a few nt can make a difference.*

We now report the data with a directly comparable RNA (24 nt) substrate (subheading “Role of helicase activity in 5’ Okazaki fragment processing”), which does not materially change our conclusions.

*5) As Dna2 must cooperate on flaps with FEN1, are there any useful insights from this structure as to how this coordination may work?*

As we discuss in the Introduction, the literature indicates that the coordination of Dna2 and Fen1 action at 5’ flaps is mediated by Rpa. As Fen1 cannot displace Rpa, it cannot act on a flap that is long enough to be coated by Rpa. This necessitates Dna2, which trims the flap until it is a poor substrate for Dna2 binding and incision as elaborated upon in our study, but an optimal substrate for Fen1 according to most published studies. As far as we know, there has been no report of a direct interaction between Dna2 and Fen1.

*Also how do the active site metal ions and bound ssDNA compare to the FEN1-DNA structure as both enzymes incise 5' flaps (Cell 2011 145(2): 198–211). What is similar and what is different in terms of the DNA binding and catalytic site?*

While both Dna2 and Fen1 incise 5’ flaps, structurally they are very distinct. Fen1 is a DNA structure-specific nuclease as it binds to the two duplexes flanking the flap and holds both the 5’ and 3’ nucleotides at the nick, while Dna2 is a single stranded DNA endonuclease that requires a free 5’ end for threading. As such, we have found no informative parallels between the two structures, beyond their use of a two-metal catalytic mechanism as in many other PD-(D/E)XK nucleases. As discussed (paragraph three, subheading “Nuclease structure and active site”), one of the two calcium ions in the Dna2 structure has a coordination shell very similar to other nucleases, while the second one differs. This is possibly due to our use of calcium instead of magnesium, as the use of mutations at either a metal ligand or the lysine in other structural studies has also been observed to cause minor shifts in metal positions.

In considering the lower activity with short 5' flaps, one wonders why is it active? The structure shows how Fen1 is controlled. Could Dna2 be "activated" somehow? Also FEN1 has significant changes upon DNA binding. For Dna2, the overall Rmsd was 0.9, but do some side chains in the nuclease have significant shifts with and without DNA?

We have not found any indications of minor conformational changes between apo and DNA-bound Dna2 that could have functional implications.

*6) Given the lowered incision activity with small ssDNA oligonucleotides, can the authors explain the 5 nt limit on 5' flaps? Is it simply that once bound, Dna2 can easily incise down to 5 nt in a processive fashion? Is it possible to distinguish the processive incision rate from an initial bind-incise rate?*

We have found no evidence for processivity using the standard assay of quenching a reaction of labeled substrate with unlabeled substrate. The 5’ flap remnant is likely too short to remain bound while the cleaved off ssDNA dissociates (or is driven off by the helicase action). *7) The displacement of RPA from the 5' end by Dna2 is interesting. However, the mechanism is difficult to understand, as the alpha1 and the OB fold are closer to where the 5' end ends up after threading than where the 5' end must thread in. Are there any added insights that can be gained from adding in the DNA-bound RPA structure, with a linker added in between 70N.*

As discussed in “Mechanism of Rpa displacement by Dna2”, the OB fold domain and alpha1 are actually closer to where the 5’ end threads into (nuclease portion of tunnel; Figure 1) than where it ends up (hel2A of helicase; Figure 1). We have clarified this by labeling the schematic and expanding the legend of Figure 6. We thank the reviewer for pointing out that the presentation of the model needs to be improved. *8) The models put forth for Dna2 action often invoke a need to thread DNA (or RNA) through the nuclease tunnel. The entirety of the tunnel would seem rather long and narrow for this to readily occur (e.g., in RecBCD/AddAB systems, movement toward the nuclease through a tunnel is aided by the helicase domains). Is it possible that the barrel and stalk might transiently undock from the body of Dna2 to make binding more favorable to the SF1B domain, and this binding could then aid threading through the (now shorter) tunnel? Also, such a conformational change occurs in the FEN1 structure where flexibility evidently aids threading prior to the formation of the active complex, so perhaps this merits a comparative comment.*

We cannot rule out the barrel and stalk transiently undocking from the bulk of the Dna2 structure, especially as it is rather mobile in the Upf1 family of RNA helicases and has high temperature factors in the Dna2 structure. However, this will not help with the initiation of threading because the tunnel entrance, which is also the narrowest part of the tunnel, is formed by the crossover loop that is stapled on the rest of the nuclease by the iron-sulfur cluster.

*9) Along similar lines, although it is true that the ATPase activity of Dna2 is not strictly essential for yeast viability, it is unclear from the model depicted in Figure 6 why Dna2 should need an ATPase domain at all. Given the conservation of this region in Dna2 homologs, this question merits some discussion/clarification.*

Our data suggests that the helicase activity helps bypass the 5’ RNA primer of Okazaki fragments. The data is described in “Role of helicase activity in 5’ Okazaki fragment processing” and shows that the helicase activity (ATP versus AMPPNP) stimulates cleavage of a mixed RNA-DNA substrate. Our structure also sheds light on the reported inhibition of 3’ end cleavage by ATP, indicating that this is due to 3’ end threading being counteracted by the helicase domain moving the DNA in the opposite direction (paragraph five, subheading “Mechanism of DNA binding”). We now reiterate this in the Conclusions:

“Instead of translocation, the helicase domain appears to be important for providing DNA affinity and for bypassing the 5’ RNA primer of Okazaki fragments.”

*10) Paragraph five, subheading “Overall structure of the Dna2-ssDNA complex”: Where does the α1 helix sit on the structure when it docks against a symmetry mate? Is this internal interaction functionally important? Would it be possible for this interaction to occur in cis?*

In the crystals, the alpha1 helix packs against the helicase 2A domain, at a site that is not conserved across species, and in an interaction that cannot occur in cis. It is also unlikely to form in solution, as we have found no evidence of dimerization at up-to ~200 micromolar Dna2. *11) Ramachandran values are not included in the Methods or in Table 1.*

We have included Ramachandran values in the Methods, subheading “Structure determination and refinement”.